# *Helicobacter pylori* and the Human Gastrointestinal Microbiota: A Multifaceted Relationship

**DOI:** 10.3390/antibiotics13070584

**Published:** 2024-06-24

**Authors:** Ege Tohumcu, Francesco Kaitsas, Ludovica Bricca, Alessandro Ruggeri, Antonio Gasbarrini, Giovanni Cammarota, Gianluca Ianiro

**Affiliations:** 1Department of Translational Medicine and Surgery, Università Cattolica del Sacro Cuore, Largo A. Gemelli 8, 00168 Rome, Italy; etohumcu09@gmail.com (E.T.); francesco.kaitsas01@icatt.it (F.K.); alessandro.ruggeri02@icatt.it (A.R.); antonio.gasbarrini@unicatt.it (A.G.); giovanni.cammarota@unicatt.it (G.C.); 2Department of Medical and Surgical Sciences, UOC Gastroenterologia, Fondazione Policlinico Universitario Agostino Gemelli IRCCS, 00168 Rome, Italy; 3Department of Medical and Surgical Sciences, UOC CEMAD Centro Malattie dell’Apparato Digerente, Medicina Interna e Gastroenterologia, Fondazione Policlinico Universitario Gemelli IRCCS, 00168 Rome, Italy; 4Department of Surgical, Oncological and Gastroenterological Sciences (DiSCOG), Padua Univeristy, 35123 Padova, Italy; ludovicabricca@hotmail.it

**Keywords:** gastritis, *Helicobacter pylori*, gut and gastric microbiota, *H. pylori* eradication therapies

## Abstract

*Helicobacter pylori* is a type of Gram-negative bacteria belonging to the Proteobacteria phylum which is known to cause gastrointestinal disorders such as gastritis and gastric ulcers. Its treatment is based on current eradication regimens, which are composed of combinations of antibiotics such as clarithromycin, metronidazole, levofloxacin and amoxicillin, often combined with a proton pump inhibitor (PPI). With the development of sequencing technologies, it has been demonstrated that not only does the colonization of the gastric and gut environment by *H. pylori* cause microbial changes, but also the treatment regimens used for its eradication have a significant altering effect on both the gastric and gut microbiota. Here, we review current knowledge on microbiota modulations of current therapies in both environments. We also summarize future perspectives regarding *H. pylori* infection, the integration of probiotics into therapy and what challenges are being faced on a global basis when we talk about eradication.

## 1. *Helicobacter pylori*: Microbiology, Epidemiology and Diagnosis

*Helicobacter pylori* (*H. pylori*) is a Gram-negative, spiral-shaped, highly motile bacterium of the phylum *Proteobacteria* [1,2]. Its genome is composed of nearly 1.6 megabase pairs (Mbps), resulting in a very small structure compared with other pathogenic bacterial species. Another distinctive feature is its high level of instability because of the uncommon pairing of exceedingly high mutation and recombination frequencies, coupled with the absence of a mismatch repair pathway and the mutagenic induction of its DNA polymerase I [3]. First discovered in 1982 by Warren and Marshall [4], it is now the most frequent cause of chronic gastritis and various other variably severe illnesses, such as peptic ulcer disease, gastric carcinoma, and gastric mucosa-associated lymphoid tissue (MALT) lymphoma [5]. Half of the global population is infected with *H. pylori*, with a higher prevalence in lower socioeconomic areas such as South America, sub-Saharan Africa and the Middle East [6], and it is mainly transmitted by fecal–oral, oral–oral and sexual routes [7].

Although most acute infections remain undiagnosed, many invasive and non-invasive diagnostic methods can be performed on suspected infections [8]. The detection of *H. pylori* relies on invasive tests such as the rapid urease test (RUT), histological analysis, bacterial culture and the detection of genetic material. The detection of genetic material is possible through fluorescence PCR in situ hybridization. Non-invasive testing methods include the urea breath test, serological testing, the stool antigen test and the detection of genetic material in stool via PCR [9].

## 2. Bacterial Elements Implicated in *Helicobacter pylori* Colonization and Pathogenesis

Due to the unfriendly gastric habitat, *H. pylori* has developed a set of pathogenic mechanisms that allow it to survive in the deep gastric mucus layer [10]. Firstly, bacterial flagella enable *H. pylori* to enter inside the gastric mucus layer, where there is a pH of 7, compared with an acidic one in the lumen, demonstrating how flagella-driven motility is an early-stage virulence factor [11]. A recent discovery has defined the role of flagellar motility inhibitors as a future treatment for *H. pylori* infection [12]. Secondly, adhesion, a well-known physical mechanism, allows *H. pylori* to adhere to the gastric mucosa and stops it from being displaced by external forces like peristalsis, epithelial cell shedding and mucus layer turnover. This mechanism is mainly mediated by blood antigen-binding protein A (BabA) and sialic acid-binding (SabA) adhesins [13]. Thirdly, *H. pylori* succeeds in regulating the gastric pH, increasing the transcription of the urease gene cluster, which is composed of seven genes [14]. One of these genes is *ureI*, which encodes for an acid-gated urea channel that opens up when the outer pH is 5, extruding a great amount of NH_3_ and NH_4_, which neutralize protons [15]. A second activity of the urease gene cluster is the production of urease enzyme, found on the *H. pylori* surface, that breaks down urea into carbon dioxide and ammonia, which, after being combined with water, produce ammonium hydroxide [16]. At last, two other pathogenic mechanisms are involved in *H. pylori* virulence: the vacuolating cytotoxin A (VacA) protein and the cag-A protein, encoded by the cagPAI complex. VacA is an oligomeric autotransporter protein that produces anion-selective membrane channels that induce apoptosis, necrosis and autophagy, resulting in a downregulation of host immune responses [17,18]. Cag-A is a pro-oncogenic protein that, after being translocated inside the host cell cytoplasm by the complex T4SS, encoded by the cagPAI complex, undergoes tyrosine phosphorylation and finally induces genome instability, nucleotide damage and the activation of the Wnt signaling pathway, which is a marker pathway in neoplasia formation [19,20].

## 3. Clinical Implications of *Helicobacter pylori* Infection 

*H. pylori* has been associated with both gastrointestinal and extra-gastric diseases [21]. Firstly, the most common illness is superficial gastritis. While acute gastritis, defined by an increase in polymorphonuclear leukocyte infiltration in the gastric mucosa, is rare, chronic gastritis, which is accompanied by an elevation in the number of lymphocytes in the mucosal layer, is the most common pathology associated with *H. pylori* [22]. Secondly, *H. pylori* infection is associated with an increased risk of developing PUD, with an odds ratio (OR) of 6.8 [23]. The pathogenesis of this disease is defined by an increase in gastrin secretion, which stimulates the release of more cloridric acid by parietal cells [24]. Thirdly, although only a small number of patients infected with *H. pylori* will present with gastric cancer, *H. pylori* infection is recognized as the primary contributor to the risk of developing gastric cancer, resulting in positive results in a minimum of 80% of gastric cancer cases [25]. The Correa cascade, a stereotyped pathological pathway, explains the development of gastric cancer [26]. In detail, a small percentage of patients with chronic gastritis develop gastric atrophy, resulting in an increased intraluminal pH that causes a reduction in somatostatin secretion and an increase in gastrin secretion, which acts as a stimulating factor for epithelial cells [27]. Then, a small amount of the population develops intestinal-type metaplasia, where CDX2 (caudal-type homeobox 2)-expressing glandular units, which are very similar to intestinal glands, replace oxyntic glands [28]. The presence of intestinal metaplasia in the stomach is associated with the presence of gastric epithelial dysplasia [29], which is linked to a risk of developing gastric cancer that is at least 10 times higher, as well as gastric extranodal marginal zone lymphomas (MALTs) [30,31]. *H. pylori* infection has been associated with extra-gastric pathologies, but evidence of these associations remains inconclusive due to limited and inconsistent data [32]. Some of them are hematologic pathologies, such as iron-deficiency anemia, idiopathic thrombocytopenic purpura and vitamin B12 deficiency [33], as well as cardiovascular disorders, ischemic heart disease, diabetes mellitus, hepatobiliary diseases, non-alcohol fatty liver disease (NAFLD) and neurodegenerative disorders [34].

The knowledge of *H. pylori* behavior and its related pathogenesis may play a relevant role in the comprehension, treatment and prevention of various related illnesses ranging from peptic ulcers to more malignant diseases, such as gastric cancer and MALT lymphoma.

## 4. The Human Gut Microbiota and Gastric Microbiota

The human gut microbiota is a superorganism which develops starting at gestational ages through the interactions of maternal microbiota and immune responses at specific niches in the body [35]. The gut microbiota, having the greatest number of microorganisms [36], is a composite network of numerous species of bacteria, fungi, viruses and archaea that reside in a common ecosystem [37]. 

In a healthy state, the intestinal microbiota is dominated primarily by *Bacteroidetes* and *Firmicutes*, followed by *Proteobacteria*, *Fusobacteria*, *Tenericutes*, *Actinobacteria* and *Verrucomicrobia*, which constitute 90% of the population [38]. The majority of the bacteria in the human gut are anaerobes, while aerobic profiles have been recorded in the cecum previously [36]. 

The composition of the gut microbiome is influenced by environmental conditions, dietary changes and other selective pressures [39]. For example, urbanization is associated with several clinical conditions, such as an increased risk of obesity, type 1 diabetes, behavioral disorders, and inflammatory bowel disease, which may have impacts on the gut microbiota as well [40,41]. On the other hand, the utilization of antibiotics, especially during infancy, has also been shown to be a major influencer of the human microbiota. For example, clarithromycin use, which is commonly utilized as a part of *H. pylori* treatment, decreases the number of Actinobacteria (the phylum that includes the beneficial Bifidobacteria) [42].

Increasing evidence suggests that the gut microbiota is involved in the pathogenesis of several disorders [43]. The increasing prevalence of obesity and its associated metabolic disorders, including type 2 diabetes and metabolic syndrome, have been associated with a state of dysbiosis [44]. Additionally, there is emerging evidence regarding the gut–brain axis and the role of the microbiota in multiple biological systems and neurological disorders, such as autism spectrum disorder, Parkinson’s disease, Alzheimer’s disease, multiple sclerosis and depression [45]. 

The gastric environment hosts its microbiome, which is less diverse than the colonic one due to the higher gastric acidity [46]. This results in a lower microbial load in the stomach (10^2^–10^4^ CFU/mL) compared to the gut (10^10^–10^12^ CFU/mL) [47]. The predominant phyla of the gastric environment include *Actinobacteria*, *Bacteroidetes*, *Firmicutes* and *Proteobacteria*, which include *H. pylori* [48,49]. 

Furthermore, recent advances have shown that in the healthy population, the gastric microbiota does not have significant differences according to geographical area and ethnicity, and it is quite similar at both the genera level and the phyla level. Meanwhile, the major driver of gastric microbiota diversity is the presence of *H. pylori* infection [47,50].

By sequencing 1833 bacterial isolates from gastric biopsies of 23 adults living in the United States, Bik et al. found that the gastric mucosa is colonized by the previously mentioned five major phyla, with the microbial populations composed of a diverse bacterial community, amounting to 128 phylotypes [49]. Also, in Llorca et al.’s study, *Proteobacteria*, *Firmicutes*, *Bacteroidetes* and *Actinobacteria* were found to be the dominant phyla in the healthy gastric microbiota, but with variable percentages of abundances [51], while other studies demonstrate that the most abundant phyla in the gastric microbiota of *H. pylori*-negative patients are *Firmicutes*, *Bacteroidetes* and *Actinobacteria* [52]. 

## 5. The Impact of *H. pylori* Infection on the Gastric Microbiota 

Being a major human gastric pathogen resident in more than 50% of the world’s population [6], *H. pylori* causes an asymptomatic infection in more than 80% of infected humans [53] but also stands as the most important risk factor for several gastric diseases, as previously mentioned [54,55]. The pathogenic mechanisms of *H. pylori* include not only a direct effect on the gastric mucosa but also indirect damage, as it disrupts the stomach microbial composition, reducing the abundance of beneficial bacteria and allowing for colonization by potentially pathogenetic bacteria [56]. Alterations which take place due to eradication therapies at both the gastric and gut levels are summarized in Table 1. 

This *H. pylori*-mediated dysbiosis in the human stomach has specific characteristics, including a significant decrease in alpha diversity compared with *H. pylori*-negative individuals [57]. Moreover, an inverse correlation between the alpha diversity of the gastric microbiome and *Helicobacter* abundance [58] characterizes *H. pylori*-mediated dysbiosis, as well as a significant decrease in the microbial beta diversity [51] and a negative interaction between both *H. pylori* and other gastric microbes [58] and among gastric microbes surviving *H. pylori* colonization [59]. 

It is also interesting to highlight how the interaction between *H. pylori* infection and the gastric microbiota is influenced by the eradication of *H pylori*.

Helicobacter eradication, indeed, can restore gastric microbiota, exerting beneficial effects on gastric microbial populations [60]. In particular, a significant increase in alpha diversity has been observed, together with an enrichment of commensal microbes, after the successful eradication of *H. pylori* [61,62]. 

Interestingly, however, it was also shown that even after eradication therapy, healthy gut commensal bacteria were not restored [63]. These findings curb the clarity of the effect of *H. pylori* eradication on the gastric microbiome. These outcomes are also time-dependent, as supported by increasing evidence suggesting that the restoration of the gastric microbiome occurs more than 2 months after eradication therapy is terminated [56,60]. Thus, whether *H. pylori* eradication therapies can also have a restoring effect on members of the gastric bacterial population remains a topic of debate [64]. 

It is relevant, furthermore, to underline that the eradication of *H. pylori* has effects not only on the composition of the gastric microbiome but also on its metabolites. In a case–control study, Peng et al. compared the metabolomic profiles of paired biopsies from patients who succeeded or failed to eradicate *H. pylori* and found that 81 metabolites related to five metabolic pathways (i.e., acylcarnitines, ceramides, triacylglycerol, cholesterol esters, fatty acids, sphingolipids, glycerophospholipids, and glycosylceramide metabolic pathways) differed significantly after successful eradication compared to failed treatment [65]. The authors also found associations between specific metabolites and taxa, i.e., Prevotella and triacylglycerols, as well as Helicobacter and cholesterol esters, ceramides and fatty acids, specifically in patients with intestinal metaplasia compared to those with superficial gastritis or chronic atrophic gastritis. The positive association between Helicobacter and fatty acids or phosphatidylcholines was shown by Dai et al. in patients with gastric cancer as well [66]. 

The idea that *H. pylori* infection status, gastric microbial diversity and the gastric juice pH value are potential factors for gastric metabolite alterations [65] is well supported by Cheung et al.’s study [67]. In particular, they established a relationship between profound acid suppression in long-term PPI users, even after *H. pylori* eradication, and an increased risk of gastric cancer. This result is likely related to the worsening of atrophic gastritis, especially in gastric atrophies established because of chronic *H. pylori*-induced inflammation [67].

Moreover, microbiome shifts associated with *H. pylori* eradication appear to also have effects on the development of precancerous lesions and gastric cancer [68,69]. Sung et al. showed that subjects who developed atrophy 1 year after *H. pylori* eradication presented a persistent emergence of atrophy and intestinal metaplasia that was associated with stomach colonization by an oral cavity bacterial cluster, affecting Peptostreptococcus, Streptococcus, Parvimonas, Prevotella, Rothia and Granulicatella [63], confirming that other organisms which engraft the stomach after *H. pylori* eradication might also be involved in gastric inflammation and gastric carcinogenesis. 

These results demonstrate that specific gastric microbiome and metabolome interactions may influence gastric inflammation and carcinogenesis, while other observations are still needed to better understand the details of this association and whether there is a causal relationship [65].

## 6. The Impact of *H. pylori* Infection on the Gut Microbiota

H. pylori infection can influence the gut microbiota through several pathways. First, the actual infection itself appears to modify the gut microbiota, as observed in several cohorts. In a large study of 212 patients with *H. pylori* infection and 212 paired controls, the authors observed an increase in alpha diversity and in the relative abundances of Prevotella, Bacteroidetes, Parasutterella, Holdemanella, Betaproteobacteria, Pseudoflavonifractor, Alisonella and Howardella in the infected patients compared to the *H. pylori*-negative controls [70]. An increase in alpha diversity metrics was also found in 70 patients with *H. pylori* infection compared with the non-infected controls (*n* = 35) [71]. Several qualitative shifts in the gut microbiota were found in another cohort of 60 *H. pylori*-infected individuals, where families of *Coriobacteriaceae*, *Enterococcaceae*, *Succinivibrio* and *Rikenellaceae*, as well as Candida glabrata and other unclassified fungi, were increased [72]. 

There is also evidence that the gut microbiota changes during the different stages of *H. pylori* infection: Juan-Juan Gao et al. assessed the fecal microbiota of 47 subjects, 24 of them with *H. pylori* infection, 23 with non-current infection, 15 *H. pylori*-negative (control group) and 8 with a past infection. In the past infection group, a decrease in *Bacteroidetes*, *Parabacteroides* and *Barnesiella* and an increase in *Firmicutes, Proteobacteria* and *Faecalibacterium* were observed compared to the normal group. In the current infection group (*n* = 24), there was also a decrease and increase in the same taxa, but in lower amounts [73]. Interestingly, a study conducted by Sun et al. including patients with *H. pylori* infection and negative controls observed that *H. pylori* infection not only causes the dysbiosis of the gut microbiota but has also been shown to be related to more severe depressive symptoms in PSD (post-stroke depression). In the *H. pylori* (+) group, there were lower amounts of *Proteobacteria* and *Verrucomicrobia*, *Akkermansia muciniphila*, *Bacteroides dorei* and *Fusobacterium ulcerans*, while *Megamonas funiformis* and *Bifidobacterium adolescentis* were more abundant in the infected group compared with the controls [74]. 

Preliminary evidence also suggests a correlation between different phases of the Correa’s cascade and the composition of the gut microbiota. In a large (*n* = 884 patients) Japanese cohort of patients with *H. pylori* infection, the severity of atrophic gastritis was positively associated with the abundance of Lactobacilli in the gut microbiota [75].

A more consolidated line of evidence suggests that the gut microbiota can also be deeply altered by *H. pylori* eradication therapies, which are mainly based on systemic antibiotics. He et al. treated 10 asymptomatic adults with *H. pylori*-related gastritis with bismuth quadruple therapy for 14 days and noticed lower alpha and beta diversity in the intestinal microbiota immediately after treatment [76]. Also, Martin-Nuñez et al. observed a lower presence of *Firmicutes*, *Actinobacteria*, *Proteobacteria* and *Verrucomicrobia* and an increase in *Bacteroidetes* after *H. pylori* eradication compared with controls. In the pre-eradication group, the same shifts were observed, but in lower amounts [77]. These differences indicate the clear influence of antibiotic treatment for *H. pylori* on the fecal composition of the patients. 

Interesting data are arising from the use of innovative eradication strategies. Specifically, vonoprazan–amoxicillin (VA) dual therapy was shown to provide only minimal and transient alterations in the gut microbiota, which were recovered only 1 month after eradication [78]. 

A long-lasting line of research has been investigating the clinical role of probiotics in *H. pylori* infection. The Maastricht guidelines for the management of *H. pylori* infection recommend adding specific probiotics to eradication therapies with the aim of decreasing the rate of side effects and therefore increasing the compliance of patients and consequently the eradication rates [79]. 

In *H. pylori*-infected patients with previous failure of *H. pylori* eradication who received probiotics (specifically Saccharomyces boulardii) during the last 2 weeks of the second attempt of eradication therapy, an increased therapy efficacy of 28.0% was shown [80]. In a study in which 247 *H. pylori*-positive patients were randomized to receive 14-day quadruple therapy (esomeprazole, bismuth, amoxicillin and furazolidone) combined with probiotics (Bifidobacterium Tetragenous viable bacteria tablets) or a placebo for 28 days, it was shown that the alteration of the gut microbiota induced by eradication regimens could be partially alleviated by probiotic treatment [81]. Thus, current studies demonstrate an additional benefit of probiotics in terms of efficacy when used in combination with Hp eradication therapies. A study led by Viazis et al. showed that twice the daily intake of probiotic supplementation significantly delayed and reduced the onset of adverse events related to the therapy [82]. There is more supportive evidence in the literature regarding how the use of probiotics alleviates the symptomatology and adverse events related to eradication therapy as well [83,84]. 

In 2017, a systematic review and meta-analysis by Wang et al. on the effect of probiotics on the rate of eradication in *H. pylori* therapy was published. This work highlighted a potential place for the insertion of probiotics into the therapy regimen for *H. pylori* infection [85]. The authors identified a total of 20,215 patients in over 140 studies. Also, in the detected studies, more than 10 probiotic strategies were used as a supplementation therapy in Hp eradication. Their study proved that probiotics could enhance efficacy and tolerance in most eradication strategies. Wang et al. also underlined that the similarity attributed to the common mechanism of probiotics could be implicated in Hp inhibition [85].

Recent important evidence on this topic comes from another network meta-analysis, comparing multiple treatment regimens in children with *Helicobacter pylori* infection, performed by Liang et al. in 2022 [86]. The study involved 163 studies, more than 18,000 children and 10 different regimens. Among these, the cumulative ranking showed that the children who used sequential therapy with probiotics in addition to antibiotics had the best outcome in terms of efficacy, while bismuth-containing regimens achieved the worst result. With this meta-analysis, it was possible to point out the higher safety profile of eradication therapies in combination with probiotics in the pediatric population as well [86]. 

## 7. Future Perspectives

Eradication therapies for *H. pylori* currently face different challenges such as antibiotic resistance, patient compliance to therapy and treatment-related side effects [87]. Increasing rates of resistance worldwide result in efficacy rates gradually declining [88]. Indeed, a meta-analysis in 2018 demonstrated resistance to antibiotics such as clarithromycin, metronidazole and levofloxacin, which are all components of current treatment regimens, exceeding high resistance rate thresholds (>15%) according to Maastricht’s consensus in many different regions [89]. The introduction of susceptibility testing through polymerase chain reaction (PCR) facilitated the concept of tailored therapy for patients, increasing the eradication rate [90]. To obtain an optimal approach, clinicians’ decision making should take into consideration the different aspects of the patient and their disease history. 

In light of the challenges in eradication and increasing side effects, the potential role of probiotics has been studied in several trials. There are several mechanisms of action supporting probiotics’ role. Suez et al. have focused on the enhancement of the gut barrier through probiotic use [91]. This effect has been proposed due to the upregulatory effect of probiotics on tight-junction proteins and increased mucin and mucus secretion in the gastric mucosa. Secondly, secretory products such as short-chain fatty acids (SCFAs), hydrogen peroxide and lactic acid are known to have an antimicrobial effect, degrading *H. pylori* [92]. Lastly, Ji and Yang et al. suggested that probiotics interfere with *H. pylori* colonization in the stomach by competing for adhesion sites and binding to *H. pylori* to facilitate its excretion [93].

As discussed in this paper, antibiotic-based therapy regimens for *H. pylori*-related gastritis are treatments that have diverse effects on the microbiome. In both the gastric and gut environments, drugs have an enhancing effect on pro-inflammatory bacteria. In addition, a decrease in alpha diversity has been shown in multiple studies as well. Given the well-known detrimental effects of continuous antibiotic use on the microbiota, the results that have been discussed are also in line with the previous literature.

With all that being said, the implementation of novel therapeutic approaches into clinical practice stands to be a main component in overcoming the increasing antibiotic resistance rates globally. Hopefully, with efficient and tailored therapy options, better disease management will also be possible.

**Table 1 antibiotics-13-00584-t001:** The outcomes of different clinical trials on the use of *H. pylori* eradication therapies and their effect on the microbiota at the gastric and gut levels.

Authors, Year	Patient Population	Outcomes	References
Fabian Frost et al., 2019	- 212 *H. pylori* (+) - 212 *H. pylori* (−)	↑ alpha diversity, ↑ *Prevotella*, ↑ *Bacteroidetes*, ↑ *Parasutterella*, ↑ *Holdemanella*, ↑ *Betaproteobacteria*, ↑ *Pseudoflavonifractor*, ↑ *Alisonella*, ↑ *Howardella*	[70]
Chen et al., 2018	- 70 *H. pylori*-positive - 35 *H. pylori*-negative	↑ alpha diversity	[71]
Nihar Ranjan Dash et al., 2019	- 60 *H. pylori* (+)	↑ *Succinivibrio*, ↑ *Coriobacteriaceae*, ↑ *Enterococcaceae*, ↑ *Rikenellaceae*, ↑ *Candida glabrata*	[72]
Juan-Juan Gao et al., 2018	- 24 *H. pylori* (+) - 15 *H. pylori* (−) - 8 with past infection	- Post-infection group: ↓ *Bacteroidetes*, ↓ *Parabacteroides*, ↓ *Barnesiella*, ↑ *Firmicutes*, ↑ *Proteobacteria*, ↑ *Faecalibacterium* - Current infection group: same shifts but lower amounts	[73]
Sun et al., 2024	Population with cerebral infarction - *H. pylori* (+) - *H. pylori* (−)	↓ *Proteobacteria,* ↓ *Verrucomicrobia*, ↓ *Akkermansia muciniphila*, ↓ *Bacteroides dorei*, ↓ *Fusobacterium ulcerans*, ↑ *Megamonas funiformis*, ↑ *Bifidobacterium adolescentis*	[74]
Iino et al., 2020	- 884 subjects with *H. pylori* infection	↑ *Lactobacilli*	[75]
He et al., 2019	- 10 asymptomatic adults with *H pylori*-related gastritis treated with BQT for 14 days - 7 age-matched adults as healthy controls	- After treatment: ↓ alpha diversity, ↓ beta diversity	[76]
Martin-Nuñez et al., 2019	- Controls - *H. pylori* (+) pre-eradication - *H. pylori* (+) treated	- After treatment: ↓ *Firmicutes*, ↓ *Actinobacteria*, ↓ *Proteobacteria*, ↓ *Verrucomicrobia*, ↑ *Proteobacteria* - Pre-eradication: same shifts but in lower amounts	[77]

## Data Availability

Not applicable.

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
