# Peer review of "Helicobacter pylori and the Human Gastrointestinal Microbiota: A Multifaceted Relationship"

_antibiotics, 2024, doi:10.3390/antibiotics13070584_

Round 1
Reviewer 1 Report
Comments and Suggestions for Authors
The topic is novel and the argument is clear.
The writing is very good.
I recommend that it can be accepted for publication in your journal.
Author Response
Thank you for your comments.
Reviewer 2 Report
Comments and Suggestions for Authors
The review article is concise and composed of all the essences required in the respective area. This can be accepted for publication with minor changes.
Current treatment strategies along with medications used for H. pylori infections can be tabulated.
In the current scenario, probiotics-based therapies are being undertaken widely for various diseases/disorders. The authors briefly discussed that in the manuscript. Additional detail on gut-microbiome in that context and the influence of H. pylori in that therapy could be added.
Concerns on the administration of antibiotics eradicating the beneficial gut microbiome highlighted, however improving health by supplementing beneficial environment to suppress H. pylori would be an ideal strategy. Evidence-based additional data on these aspects can be included.
Line 138: due to “f” the higher gastric acidity. Correct the statement.
Correct the minor changes observed throughout the manuscript. Like italics of the organism’s name, space between the genus and species name, etc.,
The initial screening has shown a 46% similarity index with published / online sources, which must be reduced to an acceptable level.
Author Response
Thank you for your valuable comments.
1. We have placed additional details about the use of probiotics in H. pylori eradication therapies that has been distributed in sections of gastric microbiota, gut microbiota and future perspectives, latter section including details about potential therapeutic outcomes.
2. We agree and acknowledge the fact that supplementing beneficial environment is a part of the eradication strategy. Several studies which have been mentioned support this strategy (e.i. use of PPIs in combination). We opted to keep the focus on implementation of probiotics when it comes to optimizing therapy.
3. line 138 has been fixed.
4. Minor changes, such as italic fonts have been modified.
5. The final screening for similarity has been reduced to an acceptable level (11%)
Reviewer 3 Report
Comments and Suggestions for Authors
The review written by Tohumcu and the coauthors deals with the determination of diversity and proportion between Helicobacter pylori and the human gastrointestinal microbiota.
The description of different role of H. pylori and its eradication, data collection and the approach to deduce the results/ conclusion are very good and interesting to readers.
Whole manuscript latin names should be italic
Whole manuscript – humans instead of people
Line 139 - (102-104 CFU/mL) compared to the gut (1010-1012 CFU/mL)
Table 1. is not cited in text
I have critical points regarding self citation of Gasbarrini A. (3 self citation as co- author) and Ianiro G, (1 self citation as co- author)
Most of the cited articles are outdated. 57 from 93 are older than 5 years.
Author Response
Thank you for your valuable comments.
1. latin names have been turned into italic
2. word "people" have been replaced with humans and/or population
3. line 139 has been fixed.
4. Table 1 has been cited in text. Thank you for your gentle reminder.
5. Prof. Gasbarrini and Prof. Ianiro have contributed in significant and solid literature works focusing on H. pylori and microbiota, specifically. Regardless of the fact that the we have worked with numerous different references, it is possible that 2 professors names are included in more than one of the papers that we have cited.
6. We have tried to use a balanced combination of both comprehensive and new literature. The trials which are older than 5 years were mostly are those with wide populations that were useful as supporting material for conclusions.